

# Technical Note: Frenkel Halsey and Hill analysis of water on clay minerals: Toward closure between cloud condensation nuclei activity and water adsorption

Courtney D. Hatch[1], Paul R. Tumminello[1], Megan A. Cassingham[1], Ann L. Greenaway[1], Rebecca Meredith[1], and Matthew J. Christie[1]

[1]Department of Chemistry, Hendrix College, 1600 Washington Ave., Conway, AR, 72032, USA

*Correspondence to*: Courtney D. Hatch (hatch@hendrix.edu)

**Abstract.** Insoluble atmospheric aerosol, such as mineral dust, has been identified as an important contributor to the cloud droplet number concentration and indirect climate effect. However, empirically-derived Frenkel-Halsey-Hill (FHH) water adsorption parameters remain the largest source of uncertainty in assessing the effect of insoluble aerosol on climate using the FHH activation theory (FHH-AT). Furthermore, previously reported FHH water adsorption parameters for illite and montmorillonite determined from water adsorption measurements below 100% RH do not satisfactorily agree with values determined from FHH-AT analysis of experimental cloud condensation nuclei (CCN) measurements under supersaturated conditions. The work reported here uses previously reported experimental water adsorption measurements for illite and montmorillonite clays (Hatch et al., 2012; Hatch et al., 2014) to show that improved analysis methods that account for the surface microstructure are necessary to obtain better agreement of FHH parameters between water adsorption and experimental CCN-derived FHH parameters.

## 1 Introduction

By mass, mineral dust is the most abundant type of aerosol in the Earth's atmosphere. The estimated average atmospheric loading of mineral dust aerosol is 19.2 Tg; nearly 63% of the total aerosol burden (Textor et al., 2006). Entrained minerals are intricately linked to climate and the hydrological cycle and have a significant impact on air quality, visibility, and health (Creamean et al., 2013; Cwiertny et al., 2008; Karanasiou et al., 2012; Mahowald et al., 2007; Prospero and Lamb, 2003; Zhu et al., 2011). The Earth's energy budget is altered due to mineral dust aerosol effects on the radiative balance of the Earth and cloud formation and lifetime (Stevens and Feingold, 2009). Nearly all, 75 to 100%, of atmospheric mineral dust emitted into the atmosphere is expected to be relatively bare minerals from arid or semi-arid source regions, not internally mixed with organic components (Forster et al., 2007; Ginoux et al., 2012). Until recently, these bare, insoluble mineral dust particles' influence on warm cloud formation has been assumed to be negligible from the lack of soluble material present (Manktelow et al., 2010; Pringle et al., 2010). However, many studies have shown that bare dust can be active cloud condensation nuclei (CCN), even if it is only weakly hygroscopic (Herich et al., 2009; Koehler et al., 2009) as CCN activity is driven by pre-adsorbed water multilayers on the surface under sub-saturated water vapor conditions (Kumar et al., 2009a; Kumar et al., 2009b; Sorjamaa and Laaksonen, 2007). Recent advances have led to an increased understanding of the role of adsorbed water on CCN activation of insoluble aerosols (Laaksonen, 2015; Laaksonen et al., 2016; Sorjamaa and Laaksonen,



2007) and regional and global models are beginning to explore their effects on climate and precipitation (Karydis et al., 2012; Karydis et al., 2011; Smoydzin et al., 2012).

Much effort has been invested in parameterizing the contribution of insoluble mineral aerosol to the number of available CCN and cloud droplets in the atmosphere. Most notably is the recent development of Frenkel-Halsey-Hill adsorption activation theory (FHH-AT) (Kumar et al., 2009a; Kumar et al., 2009b; Sorjamaa and Laaksonen, 2007). The FHH-AT framework was developed to account for the effect of adsorbed water multilayers on the CCN activity of wettable, insoluble aerosol and is based on two competing physical phenomena; the Kelvin effect and multilayer water adsorption. Since its inception, FHH-AT has been substantiated based on water adsorption, hygroscopicity, and CCN measurements of mimicked freshly-emitted mineral dust aerosol (Hatch et al., 2014; Kumar et al., 2009a; Kumar et al., 2011b, a). Additionally, recent modeling studies have incorporated the FHH-AT framework to investigate global and regional impacts of mineral dust aerosol on cloud formation (Bangert et al., 2012; Karydis et al., 2012; Karydis et al., 2011). Karydis et al. (2011) incorporated FHH-AT into the NASA Global Monitoring Initiative chemical transport model and found that insoluble mineral aerosol contributes up to 40% of the annual averaged CCN and 23% of the annual averaged cloud droplet number concentration (CDNC) in cloud-forming areas. Furthermore, dust hydrophilicity, expressed via Frenkel-Halsey-Hill (FHH) adsorption theory parameters, appears to have a more significant impact on the CDNC than dust concentration (Karydis et al., 2012). However, sensitivity studies have indicated that uncertainties in experimentally-determined FHH water adsorption parameters ($A_{FHH}$ and $B_{FHH}$) could modulate the relative contribution of mineral aerosol to cloud droplet number by up to 56% (Karydis et al., 2011). Thus, the accuracy of FHH-AT parameters affects the ability of atmospheric models to predict indirect climate effects of mineral aerosol; one of the least understood factors contributing to climate change (IPCC, 2013).

## 2 Background

Implementation of FHH-AT in climate models relies on the empirical measurement of FHH adsorption parameters obtained by applying the FHH adsorption model to experimental water adsorption measurements. The FHH adsorption isotherm describes multilayer water adsorption assuming an adsorption potential gradient based on the distance of the adsorbed water layer from the particle surface and is described by Eq. 1 (Hill, 1952):

$$s = \exp(-A_{FHH}\theta^{-B_{FHH}}) \tag{1}$$

where $s$ is the saturation ratio of water vapor above the sample, $\theta$ is the relative water coverage (or number of adsorbed monomolecular water layers) and $A_{FHH}$ and $B_{FHH}$ are FHH empirical fit parameters that describe the intermolecular interactions governing the adsorption potential. $A_{FHH}$ characterizes interactions between the surface and first adsorbed water layer and interactions between adjacent molecules and thus governs the overall extent of water coverage. Higher $A_{FHH}$ values suggest that more water can be adsorbed. $B_{FHH}$ describes the interactions between the surface and subsequent adsorbate layers. Smaller $B_{FHH}$ values characterize stronger attractive forces over greater distances from the surface. Thus, $B_{FHH}$ greatly influences the shape of the adsorption isotherm, particularly at high saturation ratios. As a result, CCN activation determined using FHH-AT is predominantly driven by the magnitude of $B_{FHH}$ (Kumar et



al., 2009a). In order to accurately determine $A_{FHH}$ and $B_{FHH}$, experimental measurements of $\theta$ as a function of relative humidity (RH) must be known to a high degree of accuracy.

FHH-AT describes the contribution of water adsorption to CCN activity by Eq. 2 (Hung et al., 2015; Kumar et al., 2009b; Sorjamaa and Laaksonen, 2007; Tang et al., 2016).

$$s = \exp(-A_{FHH}\theta^{-B_{FHH}}) \exp\left(\frac{4\sigma M_w}{RT\,_w D_p}\right) \qquad (2)$$

The first exponential term represents the effect of water adsorption in the form of the FHH adsorption model. The second exponential term represents the Kelvin effect, where $\sigma$ is the surface tension of water (7.20x10$^{-2}$ J/m$^2$) (Pruppacher and Klett, 1980), $M_w$ is the molar weight of water, $R$ is the universal gas constant, $T$ is the temperature, and $\rho_w$ is the water density. Equation 2 can be used to calculate CCN activity under supersaturated water vapor conditions if $A_{FHH}$ and $B_{FHH}$ are known based on fitting Eq. 1 to experimental water adsorption measurements at subsaturated water vapor conditions (Hatch et al., 2014; Hung et al., 2015). Alternatively, $A_{FHH}$ and $B_{FHH}$ can be determined from size-resolved experimental CCN activation measurements of the critical supersaturation, $s_c$, as a function of the dry particle diameter, $D_{dry}$ (Kumar et al., 2011b, a; Sorjamaa and Laaksonen, 2007).

Recent studies have attempted to calculate CCN activities of mineral dust components based on FHH parameters derived from experimental water adsorption parameters (Hatch et al., 2014; Hung et al., 2015). Hatch et al. (2014) found that while the calculated CCN activation was in good agreement with experimental CCN measurements of similar minerals (illite and montmorillonite clay) (Kumar et al., 2011b, a), the FHH adsorption parameters were significantly different based on the method by which they were acquired; experimental water adsorption (Hatch et al., 2014) vs. aerosol CCN activation measurements (Kumar et al., 2011b, a; Tang et al., 2016). Figure 1 shows the previously reported experimental water adsorption isotherms for (a) illite and (b) montmorillonite clays based on water adsorption measurements (Hatch et al., 2014). For comparison, Fig. 1 also shows the FHH adsorption isotherms of illite and montmorillonite based on $A_{FHH}$ and $B_{FHH}$ parameters derived from FHH-AT analysis of size-selected CCN measurements using wet (Kumar et al., 2011b) or dry (Kumar et al., 2011a) aerosol generation methods. The FHH adsorption isotherms from CCN activation measurements (dashed lines) were calculated based on reported (Kumar et al., 2011b, a) $A_{FHH}$ and $B_{FHH}$ values using Eq. 3 (Tang et al., 2016).

$$\theta = \sqrt[B_{FH}]{\frac{A_{FHH}}{-\ln(s)}} \qquad (3)$$

As shown in Fig. 1, the relative water coverage based on water adsorption measurements differs by a factor of 5 (illite) and 10 (montmorillonite) at 40% RH from the adsorption curves calculated using FHH parameters derived from aerosol CCN activation measurements. The work presented here aims to address potential sources of the outstanding differences between FHH parameters obtained from water adsorption (Hatch et al., 2014) and CCN activation (Kumar et al., 2011b, a) measurements of illite and montmorillonite clays. Previously reported experimental water adsorption measurements on montmorillonite and illite clays by Hatch et al. (2012) are used to show that improved analysis methods accounting for surface microstructure are necessary to obtain more accurate FHH adsorption parameters from water adsorption measurements and better agreement to experimental CCN-derived FHH parameters.



### 3 FHH Activation Theory Water Adsorption Analysis

The results discussed here are based on further assessment of experimental water adsorption measurements previously reported in the literature (Hatch et al., 2014; Hatch et al., 2012). Hatch et al. (2012, 2014) reported water adsorption measurements on montmorillonite (SWy-2) and illite (IMt-1) clays obtained from the Clay Minerals Society's Source

Clay Repository. Water adsorption was measured using a Horizontal Attenuated Total Reflectance Fourier Transform Infrared (HATR-FTIR) spectrometer with a humidified flow reactor. Details of experimental procedures and adsorbed water quantification can be found in the literature (Hatch et al., 2014; Hatch et al., 2012). Water content as a function of RH was reported as a mass ratio of adsorbed water to dry mineral sample mass, $m_{H_2O}/m_{sample}$ (in $g_{H_2O}/g_{sample}$), and were found to be in excellent agreement with previous gravimetric water content measurements on the same clays

(Hatch et al., 2012; Schuttlefield et al., 2007b).

Traditionally, water content reported as $m_{H_2O}/m_{sample}$ is used to determine the relative surface coverage ($\theta$) by first converting the mass ratio to an experimental coverage, $\theta_{exp}$ (molec/cm$^2$), using Eq 4 (Tang et al., 2016).

$$\theta_{exp} = \frac{m_{H_2O}}{m_{sample}} \frac{N_A}{M_{H_2O} A_{BET}} \qquad (4)$$

where $N_A$ is Avogadro's number, $M_{H_2O}$ is the molar mass of water, and $A_{BET}$ is the BET surface area using N$_2$ as the adsorbate. The relative surface coverage is then determined by dividing $\theta_{exp}$ by a maximum coverage, $\theta_{max}$, or the maximum number of water molecules per cm$^2$ that can be adsorbed to form a complete monolayer (ML) on the mineral surface. $\theta_{max}$ is often approximated as 1x10$^{15}$ molec/cm$^2$ or the inverse of the cross-sectional area of a water molecule, $1/\pi r^2$, where $r$ is the radius of a water molecule. This method for obtaining $\theta$ from the experimental mass ratio of

adsorbed water has been used in previous studies (Hatch et al., 2014; Hudson et al., 2002; Schuttlefield et al., 2007a; Schuttlefield et al., 2007b) and was recommended as the preferred technique in a recent review paper (Tang et al., 2016). However, obtaining $\theta$ in this way is based on assumptions that are not relevant to the studied systems and could introduce large uncertainties. For example, the water molecule is assumed to be spherical on a molecular scale. More significantly, $\theta_{max}$ is calculated assuming a flat surface. That is, $\theta_{max}$ represents the maximum number of

spherical water molecules that can fit on a flat surface of 1 cm$^2$. However, atmospheric mineral dust particles are widely known to exhibit significant surface microstructure and porosity leading to a significantly larger surface area than that of a flat surface. Thus, the above method for obtaining $\theta$ from a mass ratio of adsorbed water can significantly overestimate $\theta$, leading to erroneous FHH adsorption parameters upon fitting the FHH adsorption model to experimental water adsorption isotherms. The estimated $\theta_{max}$ is expected to account, at least in part, for differences in

FHH parameters and adsorption isotherms obtained from water adsorption and CCN activation measurements (Fig. 1).

More accurate $\theta$ values that account for the surface microstructure of the clay particles can be determined if the maximum ML water coverage is directly determined from experimental water adsorption data. The Brunauer Emmett

Teller (BET) adsorption model is commonly applied to multilayer adsorption isotherms to determine a sample's





specific surface area based on the amount (in volume) of adsorbate necessary to achieve ML coverage and the size of the adsorbate molecule. Equation 5 shows the linear form of the BET model (Brunauer et al., 1938).

$$\frac{\frac{P}{P_o}}{\left(1-\frac{P}{P_o}\right)V} = \frac{1}{V_m c} + \frac{(c-1)}{V_m c}\left(\frac{P}{P_o}\right) \tag{5}$$

In Eq. 5, $\frac{P}{P_o}$ represents RH, $V$ is the measured volume ($cm^3$) of surface adsorbed water, $V_m$ is the volume ($cm^3$) of water

necessary to achieve ML coverage, and $c$ is a constant that is related to the enthalpy of adsorption for any layer of adsorbed water. $V_m$ and $c$ can be determined by fitting experimental adsorption isotherms with Eq. 5 (Brunauer et al., 1938; Hatch et al., 2012). Since $V_m$ is the volume equivalent of $\theta_{max}$, the relative surface coverage can be determined by $\theta = V/V_m$ as in Hung et al. (2015). BET analysis of water adsorption on illite and montmorillonite clays showed that ML water adsorption occurs at $0.06_5\pm 0.03_2$ and $0.06_3\pm 0.03_6$ $g_{H_2O}/g_{sample}$, respectively (Hatch et al., 2012). The

volume of adsorbed water necessary to achieve ML coverage can be calculated from these ML water content values expressed as mass ratios (Hatch et al., 2012) following Eq. 6.

$$V = \frac{m_{H2O}}{m_{sample}}\frac{m_{sample}}{D_{H_2O}} \tag{6}$$

In Eq. 6, $m_{sample}$ is the mass (g) of sample, $\frac{m_{H2O}}{m_{sample}}$ represents the experimental mass ratio of adsorbed water

($g_{H_2O}/g_{sample}$), and $D_{H2}$ is the density of water at room temperature (997.045 $kg/m^3$) (Lide, 1993). Given illite and

montmorillonite sample masses of 0.8 and 0.3 mg, $V_m$ is calculated, to be $5.2x10^{-5}$ and $1.9x10^{-5}$ $cm^3$, respectively, based on BET analysis of experimental water adsorption data (Hatch et al., 2012).

For comparison, $V_m$ based on the estimated $\theta_{max}$ value of $1x10^{15}$ molec/$cm^2$ can be calculated using Eq. 7.

$$\text{Estimated } V_m = \frac{\theta_{max} M_{H_2O} A_{BET} m_{sample}}{N_A D_{H_2O}} \tag{7}$$

In Eq. 7, $M_{H_2O}$ is in kg/mol, $D_{H20}$ is in $kg/cm^3$, and $A_{BET}$ (using $N_2$ as an adsorbate) is in $cm^2/g$ (Hatch et al., 2012). A $\theta_{max}$ of $1x10^{15}$ molec/$cm^2$ is equivalent to $V_m$ values of $5.0x10^{-6}$ and $2.3x10^{-6}$ $cm^3$ water for illite and montmorillonite clays, respectively. This is approximately an order of magnitude less adsorbed water at ML coverage than $V_m$ values directly determined from experimental water adsorption data using BET analysis. Thus, previous studies that use $\theta_{max}$ to calculate $\theta$ are overestimating the relative water coverage by up to an order of magnitude. This result is consistent

with discrepancies in FHH curves determined based on previous water adsorption and CCN activation measurements illustrated in Fig. 1 and thus is likely to be a major source of the disagreement observed in the literature.

Using Eq. 6, $V$ and $\theta$, where $\theta = V/V_m$, were determined as a function of percent relative humidty based on previously reported water content mass ratios for illite and montmorillonite clays (Hatch et al., 2012). Figure 2 shows the

calculated $\theta$ for (a) illite and (b) montmorillonite as a function of percent relative humidity based on obtaining $V_m$ from BET analysis of experimental water adsorption data (Hatch et al., 2012). For comparison, adsorption curves calculated using Eq. 1 based on $A_{FHH}$ and $B_{FHH}$ parameters derived from CCN activation measurements of dry-generated illite and montmorillonite are also shown (Kumar et al., 2011b, a). In contrast to Fig. 1 which shows $\theta$ as calculated using the estimated $\theta_{max}$, Figure 2 demonstrates that direct measurement of ML water content using BET





analysis of the experimental data significantly enhances closure between adsorption isotherms derived from water adsorption and CCN activation measurements of microstructured clay minerals. As shown, the experimental adsorption curves of $\theta$ as a function of percent relative humidity are now in much better agreement with FHH adsorption curves based on FHH parameters from dry-generated illite and montmorillonite clay minerals, thus

reducing the disagreement between these two methods (Hatch et al., 2014; Kumar et al., 2011b, a; Laaksonen et al., 2016; Tang et al., 2016).

To obtain more accurate FHH adsorption parameters from the experimental water adsorption data for illite and montmorillonite clays shown in Fig. 2, Eq. 1 was rearranged to a linear relationship as shown in Eq. 8 (Tang et al.,

10    2016).

$$\ln[-\ln(s)] = \ln A_{FHH} - B_{FHH} \ln \theta \qquad\qquad\qquad (8)$$

Linear regression analysis of $\ln[-\ln(s)]$ as a function of $\ln \theta$ from 40-90% RH allows for the determination of $A_{FHH}$ and $B_{FHH}$ (Hung et al., 2015; Tang et al., 2016). According to Hung et al. (2015), constraining the FHH adsorption model fit to a limited range of high RH values, avoids uncertainties due to assumptions inherent in the FHH adsorption

theory as the fit is limited to the multilayer water adsorption regime. Importantly, the FHH adsorption model assumes that particles are spherical, of a single universal diameter, have a smooth surface and that water is uniformly distributed (Hill, 1952). These assumptions are problematic when applying to adsorption measurements on bulk, polydisperse mineral dust particles which are known to be irregularly shaped and porous. Constraining the fit to higher RH values helps avoid surface porosity effects on the resulting FHH adsorption parameters. In contrast, previous work that

reported FHH adsorption parameters for illite and montmorillonite clays was based on fitting the FHH adsorption model to the entire range of RH values studied (Hatch et al., 2014).

Figure 3 shows the FHH adsorption theory analysis of experimental water adsorption on illite and montmorillonite clays based on a constrained FHH fit as described above and $\theta$ calculated as $V/V_m$, where $V_m$ was directly measured

from the experimental water adsorption data using BET analysis. The closed circles represent the data fit to the FHH equation (Eq. 8). Resulting $A_{FHH}$ and $B_{FHH}$ values are reported in Table 1. For comparison, FHH parameters previously reported in the literature determined using other methods are also reported. FHH parameters from analysis of previously reported water adsorption data (Hatch et al., 2014) assuming $\theta = \theta_{exp}/\theta_{max}$, water adsorption (montmorillonite only) analyzed based on $\theta = V/V_m$ and the constrained FHH analysis and collected using a surface

area and porosity measurement system (Hung et al., 2015), and experimental CCN activation measurements of wet- and dry-generated clay minerals (Kumar et al., 2011b, a) are reported. In general, the FHH parameters from water adsorption measurements using the method reported here agree more closely to CCN activation-derived FHH parameters compared to those reported previously for the same sample (Hatch et al., 2014). Using Eq. 3, illite and montmorillonite water adsorption isotherms (Fig. 2, solid lines) were calculated based on $A_{FHH}$ and $B_{FHH}$ values

determined here (Table 1, this study). As shown in Fig. 2, the FHH curves based on analysis of experimental water adsorption appear to fit the experimental data very well and the adsorption isotherms show significantly improved agreement with FHH isotherms calculated from CCN activation measurements of the same clays. Although significant



advances toward closure between FHH parameters from water adsorption and CCN activation measurements are demonstrated here, differences remain between FHH parameters determined using different methods. Thus, continued efforts to identify improved agreement between FHH parameters from water adsorption and CCN activation measurements are warranted.

**4 Concluding Remarks**

Experimentally-determined FHH water adsorption parameters remain the largest source of uncertainty in assessing the role of insoluble aerosol on liquid cloud formation and the indirect climate effect (Karydis et al., 2012). Thus, accurate measurements of FHH adsorption parameters are necessary for reducing this uncertainty. However, Hatch et al. (2014) has shown that FHH parameters derived from water adsorption measurements (Eq. 1) can differ

significantly from values based on FHH-AT analysis of experimental CCN activation measurements (Eq. 2) of the same mineral components (Kumar et al., 2011b, a). The work reported here shows that improved fitting procedures and direct measurement of ML water adsorption coverage using BET analysis of experimental water adsorption data, thus accounting for surface microstructure, are necessary to obtain more accurate FHH adsorption parameters from water adsorption measurements and better agreement to experimental CCN-derived FHH parameters.

To assess the improved agreement between FHH adsorption parameters (Table 1) from water adsorption measurements and FHH-AT analysis of dry-generated mineral aerosol CCN activation measurements, a percent difference can be calculated. Results for both illite and montmorillonite clays indicate that the improved FHH analysis described here reduces the percent difference in $A_{FHH}$ from the value determined from FHH-AT assessment of dry-

generated aerosol CCN activation from 195% to ~65% difference (Hatch et al., 2014; Kumar et al., 2011a). However, the improved FHH analysis method has less of an effect on $B_{FHH}$. Agreement of $B_{FHH}$ from water adsorption and CCN activation measurements was improved for montmorillonite (from 50 % to 29% difference from dry-generated aerosol CCN measurements). In contrast, illite results show that the agreement between $B_{FHH}$ values from water adsorption and CCN activation measurements deteriorated (from 45 to 65% difference from dry-generated aerosol CCN

measurements) upon reassessment of experimental water adsorption data using the improved FHH analysis method reported here. Thus, while the improved FHH analysis significantly contributes to better agreement between $A_{FHH}$ values from water adsorption and CCN activation measurements, these improvements do little to bring closure to $B_{FHH}$ values.

As the $A_{FHH}$ value governs the overall extent of water coverage, the significant reduction in experimental $\theta$ based on more accurate measurements of ML water content from BET analysis of water adsorption is driving improved accuracy of $A_{FHH}$ values. Despite the improved agreement, the two approaches to determining FHH parameters (experimental water adsorption (Hatch et al., 2014) vs. aerosol CCN activation measurements (Kumar et al., 2011a)) still do not produce satisfactory closure. Further studies of factors that contribute to the overall shape of the adsorption

isotherm curve, such as adsorption heterogeneity, particle size, and porosity, are warranted and could lead to more accurate measurements of $B_{FHH}$ values that play a more significant role in predicting CCN activation of insoluble



aerosol particles. A recent paper (Laaksonen et al., 2016) suggests that the surface fractal dimension influences water adsorption on insoluble surfaces and thus could help achieve closure between water adsorption and CCN activation measurements. Thus, future studies will focus on the impact correcting water adsorption for the surface fractal dimension has on closure for experimentally-derived FHH adsorption parameters.

*Author contributions.* MAC, RM, and MJC contributed to data collection and RT, AG, and CH performed all associated data analysis. The manuscript was written through contributions of all authors. All authors have given approval to the final version of the manuscript.

*Data availability.* The data used in this publication is available to the community and can be accessed by request to the corresponding author.

*Competing interests.* The authors declare that they have no conflict of interest.

*Acknowledgements.* The reported study is based on work supported by the National Science Foundation under Grant# ATM-1755606, the Arkansas Space Grant Consortium NASA Training Grant #NNX10AL28H, the Hendrix College Odyssey Program, and the Hendrix College Morris and Ann Henry Odyssey Professorship.





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



**Table 1.** $A_{FHH}$ and $B_{FHH}$ determined from water adsorption and CCN activation measurements of dry and wet-generated illite and montmorillonite clays.

| Mineral Sample | Method | $A_{FHH}$ | $B_{FHH}$ | Source |
|---|---|---|---|---|
| Illite | Water adsorption ($\theta = V/V_m$ and constrained FHH fit) | 2.06 | 2.19 | This study |
| | Water adsorption ($\theta = \theta_{exp}/\theta_{max}$) | 75 | 1.77 | (Hatch et al., 2014) |
| | CCN activation (dry-generated) | 1.02 | 1.12 | (Kumar et al., 2011a) |
| | CCN activation (wet-generated) | 3.00 | 1.27 | (Kumar et al., 2011b) |
| Montmorillonite | Water adsorption ($\theta = V/V_m$ and constrained FHH fit) | 2.28 | 1.45 | This study |
| | Water adsorption ($\theta = \theta_{exp}/\theta_{max}$) | 98 | 1.79 | (Hatch et al., 2014) |
| | Water adsorption ($\theta = V/V_m$ and constrained FHH fit) | 1.25 | 1.33 | (Hung et al., 2015) |
| | CCN activation (dry-generated) | 1.23 | 1.08 | (Kumar et al., 2011a) |
| | CCN activation (wet-generated) | 0.87 | 1.00 | (Kumar et al., 2011b) |





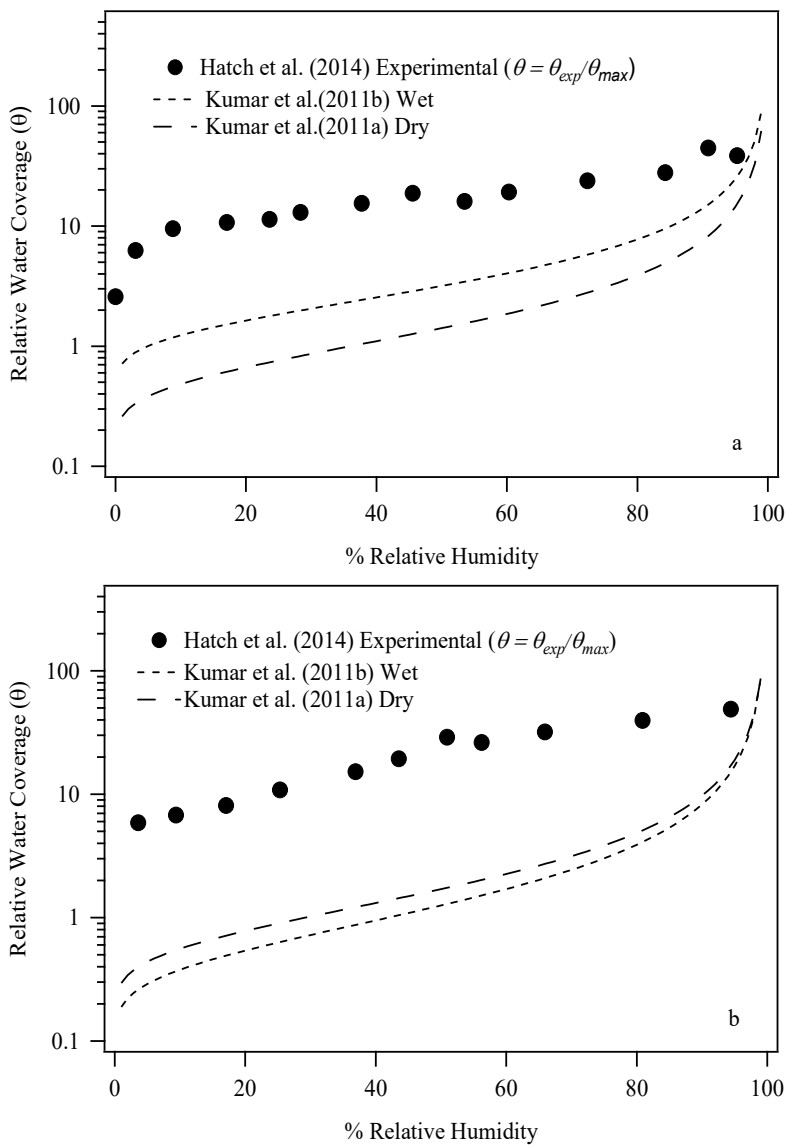

**Figure 1: Comparison of previously reported experimental water adsorption isotherms ($\theta = \theta_{exp}/\theta_{max}$) and FHH adsorption isotherms from FHH parameters determined from size-selected CCN measurements of (a) illite and (b) montmorillonite aerosol generated using wet or dry aerosol generation methods (Hatch et al., 2014;Kumar et al., 2011b, a).**





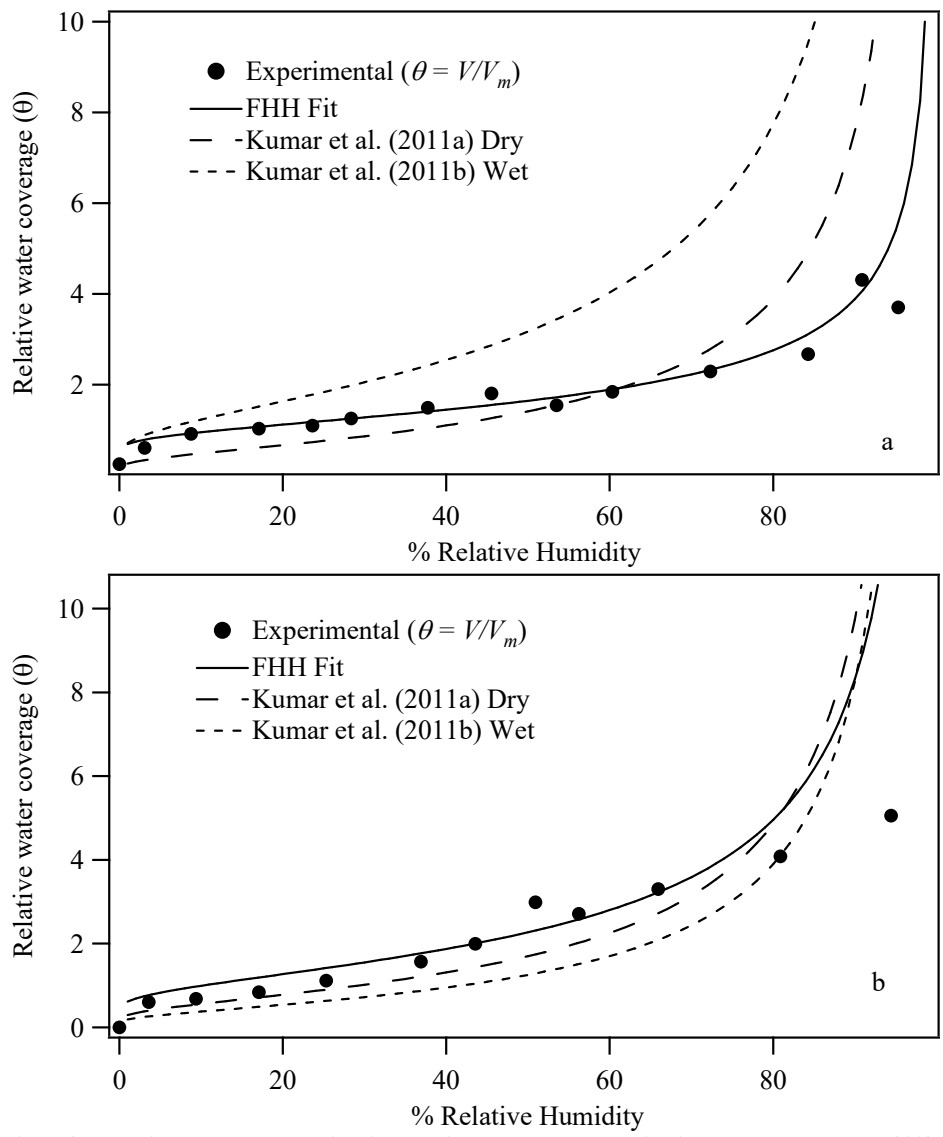

**Figure 2:** **(a) Illite and (b) montmorillonite experimental water adsorption isotherms (Hatch et al., 2012) and associated FHH fit based on constrained FHH analysis, where the experimental $\theta$ was calculated as $V/V_m$ and $V_m$ was determined from BET analysis of adsorption isotherm. FHH adsorption isotherms from FHH parameters determined from size-selected CCN measurements of aerosol generated using wet or dry aerosol generation methods are also shown (Kumar et al., 2011a).**



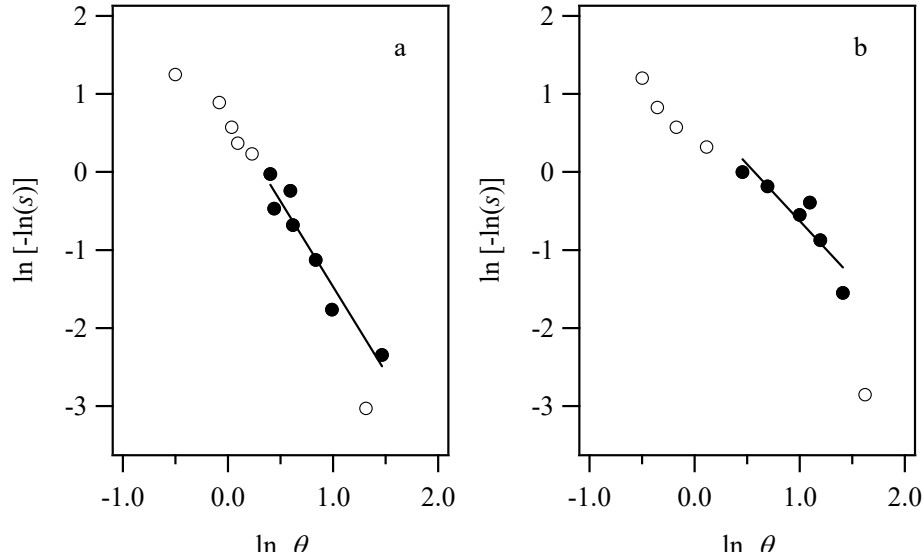

**Figure 3: FHH analysis of experimental (a) illite and (b) montmorillonite water adsorption data in which $\theta$ was calculated as $V/V_m$ and $V_m$ was determined from BET analysis. All data (open circles) are shown and the region of the constrained FHH fit (line) is represented by closed circles from 40-90% RH.**

