# Peer review of "Technical Note: Frenkel Halsey and Hill analysis of water on clay minerals: Toward closure between cloud condensation nuclei activity and water adsorption"

_Atmospheric Chemistry and Physics, 2019_

## Referee Comment (RC1) · Anonymous Referee #2 · 18 Jun 2019

General Comments: Manuscript by Hatch et. al. focuses on improved fitting procedures to obtain better agreement between FHH parameters from water adsorption measurements and CNN-derived FHH parameters. Given the importance to the CNN activity and role of adsorbed water in atmospheric chemical and photochemical processes, the information discussed in this article has become vital, more than ever, to explain field and laboratory observations and extract reaction mechanisms. Authors present scientific results from an improved parameterization of adsorbed water using a direct measurement of monolayer water coverage, determined by the BET analysis of experimental water adsorption. This approach shows a significant reduction of the gap in the FHH parameters between water uptake and CNN activity. This study is also significant due to the fact that it highlights the others potential sources for the un-modelled variations. The knowledge presented in this work will be greatly useful for atmospheric studies, in particular for better understanding heterogeneous processes on mineral dust under humid conditions. Considering all the above facts, I recommend that this article is on Langmuir after addressing the following minor concerns.

Specific Comments: Can these calculations be applied to oxide mineral dusts? If so, will the assumptions have made for clay minerals applicable?

Technical Comments: Reference style is not consistent. P4, L2 and elsewhere.

---

## Referee Comment (RC2) · Ari Laaksonen (Referee) · 24 Jul 2019

The subject of this paper is reconciliation of water vapor adsorption and CCN activation measurements of clay minerals in terms of the FHH adsorption theory. The subject matter of the paper is important, and I recommend publication, in spite of the fact that the main message of the paper – that the FHH parameters should be determined by fitting the theory to the multilayer portion of the adsorption isotherm, rather than to the complete isotherm – is rather trivial. However, I have some issuess that I believe should be discussed in the paper.

[Figure]

First, I am not completely convinced about the correctness of the FHH fitting to the montmorillonite data. Montmorillonite swells as RH is increased, up to 72% RH (Cases et al., 1992). In other words, the true multilayer portion of the adsorption isotherm occurs at higher relative humidities. In Laaksonen et al (Sci. Rep. 2016), the fitting was done to the high RH portion of the data of Hung et al (2015). The FHH plot can be seen in the supplement of Laaksonen et al, and there is a clear change of slope at around 70% RH. Also, the FHH plot of the data of Mooney et al. (JACS, 1952) shows a similar (in fact, even clearer) change. The present data is somewhat noisy, and there are only two data points at sufficiently high RH, so I understand that fitting to those two data points would not be feasible. But the matter should definitely be discussed.

Secondly, the surface fractal dimension (D) approach of Laaksonen et al. (Sci. Rep. 2016) is mentioned briefly in the end of the paper. Laaksonen et al. gave D-values for illite based on two different techniques that make use of nitrogen adsorption. Applying those D-values to the present B-parameter of illite would lead to corrected B values that are between 0.7-1.3 (I don't think there is much point to apply the montmorillonite D-values to the present data as the FHH fit is so uncertain). It should, however, be kept in mind that the data used in Laaksonen et al. (2016) was based on clays from different sources than in the current paper, and the clays may have been heat treated before the measurements, which can influence the D-values. Therefore it would be ideal if the D values could be calculated from the BET analyzer measurements mentioned in Hatch et al (2012). In any case I would suggest expanding the discussion related to the surface fractal dimension somewhat.

---

## Author Comment (AC1) · 3 Sep 2019

Author Response to Referee #2:

The *referee's comments are bolded and italicized* while our comments are in plain text

*Manuscript by Hatch et. al. focuses on improved fitting procedures to obtain better agreement between FHH parameters from water adsorption measurements and CNN-derived FHH parameters. Given the importance to the CNN activity and role of adsorbed water in atmospheric chemical and photochemical processes, the information discussed in this article has become vital, more than ever, to explain field and laboratory observations and extract reaction mechanisms. Authors present scientific results from an improved parameterization of adsorbed water using a direct measurement of monolayer water coverage, determined by the BET analysis of experimental water adsorption. This approach shows a significant reduction of the gap in the FHH parameters between water uptake and CNN activity. This study is also significant due to the fact that it highlights the others potential sources for the un-modelled variations. The knowledge presented in this work will be greatly useful for atmospheric studies, in particular for better understanding heterogeneous processes on mineral dust under humid conditions. Considering all the above facts, I recommend that this article is on Langmuir after addressing the following minor concerns.*

The authors would first like to thank Referee #2 for their assessment of the submitted work. In particular, highlighting the importance of reducing assumptions in experimental variables that can influence the modeled outcomes, for this is a major finding that challenges the current literature (Tang et al., 2016).

*Specific Comments:*
1. *Can these calculations be applied to oxide mineral dusts? If so, will the assumptions have made for clay minerals applicable?*

Indeed, the detailed analysis of water adsorption, including BET and FHH adsorption fitting methods, are applicable to oxide mineral dusts as long as multilayer water adsorption (Type II adsorption isotherms) is observed. One caveat is that the fit constraints rely on the range of RH values in which monolayer and multilayer adsorption occurs, particularly in the case of the FHH fit. The authors find that the range of RH values over which the FHH adsorption model is constrained must represent the multilayer adsorption regime. As indicated in a separate comment by Ari Laaksonen, the range of RH values over which the FHH adsorption model is fit for the montmorillonite clay may be too large as montmorillonite tends to begin multilayer water adsorption at ~70% RH. However, due to the limited number of data points available in that high RH range, we have kept the constraints on the FHH fit as is standard procedure (40-90% RH) but added a detailed description of how this might add uncertainty to the FHH results for montmorillonite. Specifically, we have added the following statement to pg. 6 of the revised manuscript:

> "However, swelling clay minerals, such as montmorillonite, are problematic as the multilayer adsorption regime begins at higher RH values. For example, previous studies have shown that the multilayer adsorption regime begins at ~70% RH for montmorillonite clay (Cases et al., 1992; Mooney et al., 1952), and thus the FHH fit should be constrained from 70-90% RH. Unfortunately, the limited number of data at high RH values precludes

the feasibility of fitting the FHH model over this smaller range of RH values. Thus, the FHH fit parameters reported here for montmorillonite suffer from uncertainty due to the swelling action of smectite clays."

This is not the case for non-swelling clays and oxide mineral dusts which typically show multilayer adsorption between 40-90% RH.

***Technical Comments: Reference style is not consistent. P4, L2 and elsewhere.***

The revised document has been edited for consistency in referencing style. Thank you for bringing awareness to this issue.

Author Response to Ari Laaksonen:

The referee's comments are bolded and italicized while our comments are in plain text

***The subject of this paper is reconciliation of water vapor adsorption and CCN activation measurements of clay minerals in terms of the FHH adsorption theory. The subject matter of the paper is important, and I recommend publication, in spite of the fact that the main message of the paper – that the FHH parameters should be determined by fitting the theory to the multilayer portion of the adsorption isotherm, rather than to the complete isotherm – is rather trivial. However, I have some issues that I believe should be discussed in the paper.***

First, the authors would like to thank Ari Laaksonen for the thoughtful assessment of our manuscript and comments that have greatly improved the quality and clarity of the work. In an effort to provide clarity, indeed constraining the FHH adsorption fit to the multilayer adsorption regime is trivial and thus resulted in trivial improvements to the overall analysis, but improvements nonetheless. However, the constraints were only one of the adjustments to the adsorption analysis. The most significant improvement observed was due to the direct measurement of the monolayer water content on the sample based on BET water adsorption analysis. Rather than the assumptions of ML coverage on a flat surface from previous analyses, the direct measurement of the ML water content on the specific sample studied showed an order of magnitude improvement in the relative water coverage compared to previous measurements (see difference between Fig. 1 and Fig. 2). This change resulted in a reduction of the percent difference in the $A_{FHH}$ parameter from 195% to 65% between the value obtained from water adsorption and CCN activation measurements, which is quite significant, and more of an effect than constraining the FHH adsorption fits. In an effort to clarify this misconception, we have revised the concluding remarks on Pg. 8, 2$^{nd}$ paragraph to include the following statement:

> "Thus, as the $A_{FHH}$ value governs the overall extent of water coverage, the significant reduction in experimental $\theta$ based on direct measurements of ML water content from BET analysis of water adsorption is driving improved accuracy of $A_{FHH}$ values."

***First, I am not completely convinced about the correctness of the FHH fitting to the montmorillonite data. Montmorillonite swells as RH is increased, up to 72% RH (Cases et al., 1992). In other words, the true multilayer portion of the adsorption isotherm occurs at higher relative humidities. In Laaksonen et al (Sci. Rep. 2016), the fitting was done to the high RH portion of the data of Hung et al (2015). The FHH plot can be seen in the supplement of Laaksonen et al, and there is a clear change of slope at around 70% RH. Also, the FHH plot of the data of Mooney et al. (JACS, 1952) shows a similar (in fact, even clearer) change. The present data is somewhat noisy, and there are only two data points at sufficiently high RH, so I understand that fitting to those two data points would not be feasible. But the matter should definitely be discussed.***

The reviewer makes an excellent point here. Indeed the FHH fit to the montmorillonite isotherm stretches outside the range of the multilayer regime and thus the resulting FHH adsorption parameters may be exhibit uncertainty due to the selected RH range used. The use of the standard range of 40-90% RH was indeed selected due to the limited number of data points at high RH values. The authors have now included the following statements, which will clarify the

uncertainty associated with choosing FHH fit constraints, and the necessity for fitting the FHH adsorption model over the multilayer adsorption regime, which may vary from sample to sample.

> "However, swelling clay minerals, such as montmorillonite, are problematic as the multilayer adsorption regime begins at higher RH values.  For example, previous studies have shown that the multilayer adsorption regime begins at ~70% RH for montmorillonite clay (Cases et al., 1992; Mooney et al., 1952), and thus the FHH fit should be constrained from 70-90% RH.  Unfortunately, the limited number of data at high RH values precludes the feasibility of fitting the FHH model over this smaller range of RH values.  Thus, the FHH fit parameters reported here for montmorillonite suffer from uncertainty due to the swelling action of smectite clays."

***Secondly, the surface fractal dimension (D) approach of Laaksonen et al. (Sci. Rep. 2016) is mentioned briefly in the end of the paper. Laaksonen et al. gave D-values for illite based on two different techniques that make use of nitrogen adsorption. Applying those D-values to the present B-parameter of illite would lead to corrected B values that are between 0.7-1.3 (I don't think there is much point to apply the montmorillonite D-values to the present data as the FHH fit is so uncertain). It should, however, be kept in mind that the data used in Laaksonen et al. (2016) was based on clays from different sources than in the current paper, and the clays may have been heat treated before the measurements, which can influence the D-values. Therefore, it would be ideal if the D values could be calculated from the BET analyzer measurements mentioned in Hatch et al (2012). In any case I would suggest expanding the discussion related to the surface fractal dimension somewhat.***

The authors would like to thank Ari Laaksonen for a great suggestion that has greatly improved the significance of the submitted manuscript.  As suggested, the authors have now added an additional discussion (beginning on pg 7 of the revised manuscript) and analysis using the fractal FHH adsorption model specifically for the illite clay.  Based on the results of this analysis, the combined effects of

[revised manuscript text omitted]

The results of this analysis are shown above in a new **Fig. 4** of the revised document. Additionally, the second paragraph of the concluding remarks has been updated based on the new results from application of the fractal FHH adsorption model.

Unfortunately, the $N_2$ BET adsorption measurements for these particular samples were outsourced due to limited resources of our institution and thus the authors do not have, nor do we have access to, the raw BET adsorption isotherms for these specific samples. Thus, the surface fractal dimension was determined directly from the water adsorption measurements, and were found to be consistent with previously reported values, although we have added a note with regard to the differences in illite samples studied.